# Stk10 Deficiency in Mice Promotes Tumor Growth by Dysregulating the Tumor Microenvironment

**DOI:** 10.3390/biology11111668

**Published:** 2022-11-15

**Authors:** Jin-Xia Ma, Dan-Dan Xu, Shun-Yuan Lu, Qian-Lan Wang, Lu Zhang, Rui Guo, Ling-Yun Tang, Yan Shen, Chun-Ling Shen, Jin-Jin Wang, Li-Ming Lu, Zhu-Gang Wang, Hong-Xin Zhang

**Affiliations:** 1Research Center for Experimental Medicine, State Key Laboratory of Medical Genomics, Shanghai Ruijin Hospital, Shanghai Jiao Tong University School of Medicine, Shanghai 200025, China; 2Shanghai Model Organisms Center, Shanghai 201321, China; 3Shanghai Institute of Immunology, School of Medicine, Shanghai Jiao Tong University, Shanghai 200025, China

**Keywords:** Stk10, tumor microenvironment, CTL, angiogenesis

## Abstract

**Simple Summary:**

The tumor microenvironment (TME) is a highly complex biological ecosystem which plays critical roles in cancer growth, evolution, and therapeutic efficacy. Identification of novel TME modulators is helpful to find new candidate targets for diagnostics and therapeutics of malignant tumors. The object of this study was to evaluate the role of serine-threonine kinase 10 (STK10) in the TME and host anti-tumor response. Our data indicate that the expression of STK10 is significantly positively associated with tumor-infiltrated immune cells. Further in vivo data revealed that Stk10 participates in anti-tumor response by regulating the activated tumor-infiltrated CTLs and tumor angiogenesis. Collectively, this is the first attempt to evaluate the correlation between STK10 and host anti-tumor response. Our data provide us with the possibility of using STK10 as a candidate target for anti-tumor immunotherapy.

**Abstract:**

Serine-threonine kinase 10 (STK10) is a member of the STE20/p21-activated kinase (PAK) family and is predominantly expressed in immune organs. Our previous reports suggested that STK10 participates in the growth and metastasis of prostate cancer via in vitro and in vivo data. However, the correlation between STK10 and the tumor microenvironment (TME) remains unclear. In this study, we assessed the relationship between STK10 and the immune cells in the tumor microenvironment of prostate cancer through bioinformatic analysis, and investigated the role of Stk10 in tumor growth using an *Stk10* knockout mouse model. The results showed that STK10 is significantly associated with the tumor-infiltrating immune cells including lymphocytes, neutrophils, macrophages and dendritic cells. The target deletion of host *Stk10* results in increased tumor growth, due to decreased activated/effector cytotoxic T lymphocytes (CTLs) and increased vessel density in the TME. In conclusion, we demonstrate that host Stk10 is involved in the host anti-tumor response by modulating the activated tumor-infiltrated CTLs and angiogenesis.

## 1. Introduction

Serine-threonine kinase 10 (STK10), also called lymphocyte-oriented kinase (LOK), is a typical member of the germinal center-like kinase (GCK)-V subfamily of the STE20/p21-activated kinase (PAK) family [1]. STK10 is mainly localized in the cellular membrane protrusions and phosphorylates several target proteins such as PLK1 (polo-like kinase 1) and ERM (Ezrin, Radixin, and Moesin) proteins [2,3]. ERM proteins have been identified as critical regulators of cell migration and invasion. Studies have reported that STK10 is involved in the pathogenesis of several cancers with diverse pathologies in a cell-specific manner [4,5,6]. Our previous work also explored the roles of STK10 in different types of cancer cells. We found that targeted deletion of STK10 in prostate cancer cells could promote cell proliferation and inhibit cell apoptosis via inhibiting the activity of p38 MAPK, and that it suppressed cell migration through inhibiting ERM proteins activation [7]. In contrast, depletion of STK10 in cervical cancer cells could promote the cell migration, invasion and adhesion in an ERM-independent manner [8]. Together, observations suggest that Stk10 executes various physiological functions by regulating diverse signaling pathways, depending on the tissue.

According to the data from Human Protein Atlas, STK10 is broadly expressed in immune cells such as NK cells, dendritic cells and T cells, which suggests that STK10 might play a critical role in the development or function of the immune system (https://www.proteinatlas.org/ENSG00000072786-STK10#gene_information, accessed on 14 August 2022) [9]. In fact, STK10 has been reported to be involved in the chemokine-induced lymphocyte migration and polarization, and LFA-1-mediated adhesion of lymphocytes [3,10]. Recently, a bioinformatics analysis by Bi et al. showed that STK10 was associated with tumor-infiltrating lymphocytes in acute myeloid leukemia [6]. However, the impact of STK10 on the host defense against tumors remains unclear, and the biological function of STK10 in the homeostasis of the tumor microenvironment (TME) needs to be further elaborated. The TME is a complex network comprised of malignant cells, infiltrating immune cells, tumor-related fibroblasts, endothelial cells, stromal cells, vascular network, extracellular matrix, and some other non-cellular components such as cytokines and chemokines [11,12,13,14]. Being an environment tumor cells reside in, the TME plays critical roles in the development, clinical outcomes and even treatment responses of malignant tumors. Any change in the composition of the TME will affect the generation and progression of tumors. A better understanding of the contributions of these factors to cancer hallmarks would be helpful to improve the current diagnostic and therapeutic strategies for malignant tumors [15,16].

To investigate the potential physiological role of STK10 in the anti-tumor immune response, we performed an integrative analysis of the influence of *Stk10* knockout in the host on tumor growth and the components of the tumor microenvironment, by using a global *Stk10* knockout mouse model and a syngeneic RM-1 murine prostate tumor cell line on the C57BL/6J background. Our data revealed that the absence of host Stk10 results in rapidly growing tumors. Further data showed that tumors grown in *Stk10* knockout mice contain low levels of activated CD8^+^ cytotoxic T lymphocytes (CTLs) and high levels of vasculature. Our data indicate that host Stk10 participates in the anti-tumor process via modulating the activated tumor-infiltrated CTLs and angiogenesis in the TME.

## 2. Material and Methods

### 2.1. Tumor Cell Lines and Mice

The RM-1 murine prostate cancer cell line (ATCC CRL-330) and HUVEC cells (ATCC PCS-100-010) were obtained from ATCC. Both cell lines were cultured in Dulbecco’s modified eagle medium (Hyclone Laboratories, Inc., Logan, UT, USA) supplemented with 10% (*v*/*v*) fetal bovine serum (Gibco, Thermo Fisher Scientific, Inc., Waltham, MA, USA) at 37 °C in a 5% CO_2_ incubator.

*Stk10* knockout mice were developed at Shanghai Model Organisms Center, Inc. (Shanghai, China) using CRISPR/Cas9 mediated gene editing technology. In brief, Cas9 mRNA was transcribed in vitro by using the mMESSAGE mMACHINE™ T7 ULTRA Tra nscription Kit (Thermo Fisher Scientific, Inc., Waltham, MA, USA) following the manufacturer’s instructions, and linearized using restriction enzyme Not I (NEB, Ipswich, MA, USA). It was subsequently purified with MEGAclear^TM^ Kit (Thermo Fisher Scientific, Inc., Waltham, MA, USA). The two target sites of the single-guide RNA (sgRNA) for exon 4 of murine *Stk10* gene were as follows: GAGGCTCTCAACTTCCTGCA and ATCCACCGAGACCTGAAAGC. The sgRNAs were transcribed in vitro using the MEGAshortscript^TM^ T7 Transcription Kit (Thermo Fisher Scientific, Inc., Waltham, MA, USA) and purified with MEGAclear^TM^ Kit (Thermo Fisher Scientific, Inc., Waltham, MA, USA). Then, the in vitro transcribed Cas9 and sgRNAs were co-injected into C57BL/6J mouse zygotes. F0 founder mice were validated by PCR and sequencing using primer pairs Stk10-test-F1 and Stk10-test-R1, which are listed in Appendix A. The positive F0 mice were further crossed with wild-type C57BL/6J mice to generate F1 mice. F1 mice with 22 bp deletions in exon 4, which produced a frameshift mutation in *Stk10*, were chosen to intercross to produce WT (*Stk10^+^*^/*+*^), heterozygous (*Stk10^+^*^/−^), and homozygous (*Stk10*^−/−^) littermates for further study. Mice were housed in individual ventilated cages under specific pathogen-free conditions in the animal facility of the Research Center for Experimental Medicine of Ruijin hospital with adequate food. Then, 6–8-week-old male Stk10 knockout homozygous (*Stk10*^−/−^) and WT (*Stk10^+^*^/*+*^) C57BL/6J mice were used in further analyses. All the protocols of animal experiments were approved by the Animal Ethics Committee of Ruijin Hospital Affiliated with Shanghai Jiao Tong University School of Medicine (Shanghai, China).

### 2.2. Bioinformatic Analysis

The expression data of STK10 protein were collected and analyzed from The Human Protein Atlas database (https://www.proteinatlas.org/, accessed on 14 August 2022), which contains the expression profile information across various tissues by immunohistochemistry (IHC) staining. The correlation between STK10 expression and immune cells infiltrated in prostate adenocarcinoma (PRAD) was analyzed on the TIMER 2.0 online database (Tumor Immune Estimation Resource, https://cistrome.shinyapps.io/timer/, accessed on 14 August 2022). The correlations with STK10 expression in 24 kinds of infiltrated immune cells in PRAD samples were analyzed using Sanger-box (http://vip.sangerbox.com/, accessed on 14 August 2022) and TCGA dataset. The expression level of STK10 mRNA in tumors and adjacent normal tissues, and the stromal score, were analyzed on a website (https://www.xiantao.love/, accessed on 14 August 2022) based on TCGA and GTEx databases.

### 2.3. RNA Extraction and qRT-PCR Analyses

Total RNAs of different mouse tissues were extracted by using the RNeasy Plus Mini Kit (Qiagen, Dusseldorf, Germany) and subsequently reverse-transcribed with AMV Reverse Transcriptase (Takara, Dalian, China) according to the instructions from the manufacturer. Then, quantitative real-time polymerase chain reaction (qRT-PCR) analyses using FastStart Universal SYBR Green Master (Roche, Basel, Switzerland) were performed on the Mastercycler ep realplex instrument (Eppendorf, Hamburg, Germany). Relative expression levels of murine Stk10 in mouse tissues were normalized to the expression of GAPDH and further analyzed using the 2^−ΔΔCt^ method. All of the primers are listed in Appendix A.

### 2.4. Label of RM-1 Cells with Luciferase

Lentivirus loading a firefly Luc gene were transfected into RM-1 cells in a 6 cm dish. After 12 h of transfection, the medium was replaced with fresh medium. Three days post-infection, the cells were cultured in complete medium with 4 μg/mL puromycin for 1 week. After characterizing the luciferase expression by using the IVIS system (IVIS-Lumina series Ⅲ, Norwalk, CT, USA), several single cell clones of RM-1-luc were identified for further study.

### 2.5. In Vivo Tumorigenicity Assay

*Stk10*^−/−^ and wild-type C57BL/6J mice 6–8 weeks of age, weighing about 20 g, were subcutaneously injected with RM-1-luc cells into the shaved right flank area for a total volume of 100 μL (1 × 10^6^ cells). After nine days of inoculation, tumor growth was measured by using a digital caliper once a day. The tumor volume was calculated using the equation as follows: *(length* × *width*^2^*)*/2. On day 17 after implantation, all of the mice were euthanized with CO_2_.

### 2.6. In Vivo Bioluminescence Assay

After 7–8 min after intraperitoneal injection with luciferin (150 mg·kg^−1^), RM-1-luc tumor-bearing mice were anesthetized with isoflurane. Then, the in vivo imaging was performed with an IVIS-Lumina series Ⅲ imaging system and subsequently analyzed with Living Image^®^ software, version 3.0.4 (Xenogen, Hopkinton, MA, USA). The exposure time was set as 1 min. Total flux of the ROI was collected as photons/s per animal. In all analyses, background luminescence on each image was normalized with an additional ROI.

### 2.7. Immunohistochemistry (IHC) and Immunofluorescence (IF) Staining

The tumor tissues were fixed in paraformaldehyde (4%) at room temperature for 48 h and then embedded in paraffin. Sections of 5 µm in thickness were cut for further analyses. The sections were deparaffinized, rehydrated, antigen retrieved and subsequently blocked with blocking buffer (10% goat serum, 0.5% Triton X-100 in 1×PBS). For IHC, the slides were first incubated with primary antibodies against indicated primary antibodies at 4 °C overnight, and then incubated with HRP-conjugated anti-rabbit antibody at room temperature for 1 h. Then, the sections were incubated with DAB and further counterstained with hematoxylin. For IF, the slides were first incubated with primary antibodies against CD31 at 4 °C overnight and then incubated with Alexa Flour 488 goat anti-rabbit IgG (H + L) (1:500, A11008, Invitrogen, Waltham, MA, USA) and DAPI (1 μg/mL, D9542, Sigma, Livonia, MI, USA) at room temperature for 2 h. Images were taken by using a microscope (Nikon, Tokyo, Japan), and the staining intensity was quantified with Image Pro Plus 6.0 software (Media Cybernetics, Inc., Rockville, MD, USA). Details of all antibodies are listed in Appendix A.

### 2.8. Flow Cytometry Analysis

Fresh tumor tissues were washed with ice-cold PBS and further washed with ice-cold RPMI 1640 medium. Then, the samples were moved into aseptic utensils and cut into small, approximately 1 mm^3^, pieces, and subsequently incubated with digest buffer (0.5 mg/mL DNase I and 1 mg/mL collagenase type I in RPMI 1640 complete medium) at 37 °C for 30 min and filtered by using a 70 μm nylon mesh. RBCs were removed by using RBC lysis buffer (BD Bioscience, New York, NY, USA). To further examine the percentages of immune cells, the single cells were incubated with the following mouse-specific antibodies—CD45, B220, CD3, CD8, CD69, CD44, CD62L, PD-1, CTLA4, CD11b, NK1.1, Gr-1, F4/80 and 7AAD—in the dark at 4 °C for 30 min at 4 °C. The apoptosis of CD8^+^ T cells was determined by using an Annexin V apoptosis detection Kit (Thermo Fisher Scientific, Inc., Waltham, MA, USA) according to the manufacturer’s instructions. Then, samples were analyzed on a FACSVerse flow cytometer (BD Bioscience, Franklin Lake, NJ, USA), and data were processed by using FlowJo software (BD Bioscience, Franklin Lake, NY, USA). Details of all antibodies are listed in Appendix A.

### 2.9. Western Blot Analysis

Mouse splenocytes and cultured cells were lysed using RIPA buffer in the presence of protease inhibitor and phosphatase inhibitor. The BCA protein assay was used to quantify the protein concentrations. Then, the protein samples were separated by denaturing 10% SDS-PAGE (sodium dodecyl sulfate polyacrylamide gel electrophoresis) at 70 V for 3 h. The proteins were transferred to a nitrocellulose filter membrane (Amersham, Little Chalfont, UK) at 250 mA for 2 h. The membranes were incubated with the block buffer (5% skim milk in a 0.1% tween-PBS solution) at room temperature for 1 h, and subsequently incubated with the primary antibodies overnight. The membrane was washed using PBST and PBS, followed by incubation of RDyeCW800-conjugated secondary antibodies for 1 h at room temperature and visualization using the LI-COR Odyssey imaging system (LI-COR, Lincoln, NE, USA).

### 2.10. Tube-Formation Assay

In vitro angiogenesis analysis was performed using Matrigel matrix (BD Biosciences, NY, USA). Chilled Matrigel matrix was added onto an ibidi culture plate (10 μL/well) and incubated for solidification at 37 °C in a 5% CO_2_ incubator. HUVECs and the RM-1 tumor cell line were separately cultured in DMEM with 10% serum. Then, the culture medium of RM-1 cells was removed, and the cells were cultured in serum-free DMEM medium for 24 h. The supernatant was collected and filtered using an ultrafiltration device (Millipore, Waltham, MA, USA). Then, 5000 HUVECs were resuspended with 50 µL of tumor cell culture medium supernatant and planted onto the gel. They were incubated for 4–6 h and then observed using an inverted microscope. Meanwhile, the number of the branching points per field was counted and calculated by using Image J software (version 1.8.0.112, Media Cybernetics, Inc., Rockville, MD, USA)

### 2.11. Statistical Analysis

All the data are presented as the mean ± standard error (*n* ≥ 3). Student’s *t* test was used to evaluate the statistical differences between groups. The statistical significance threshold was set as *p* < 0.05. The statistical graphing was performed by GraphPad Prism 7.00 software (GraphPad Software, San Diego, CA, USA).

## 3. Results

### 3.1. STK10 Was Associated with the Infiltrated Immune Cells in Prostate Cancer

To elucidate the expression pattern of STK10, we collected and analyzed the protein expression profile information of various tissues by IHC from the Human Protein Atlas database, and found that STK10 is dominantly expressed in immune organs, such as the spleen and lymph nodes (Appendix A). The results indicated that STK10 might participate in the development of the host immune system and/or immune responses. In our previous study, we revealed that STK10 participates in the pathogenesis of prostate cancer, but we did not evaluate whether STK10 is involved in the host’s anti-tumor immune response. To address this question, herein, we further evaluated the potential role of Stk10 in the anti-tumor response by analyzing the association between STK10 and the infiltrated immune cells of prostate adenocarcinoma. According to the data from the TIMER database, the expression of STK10 showed significant association with lymphocytes, neutrophils, macrophages and dendritic cells in PRAD tumor tissues (Figure 1A). Subsequently, we tested the expression of STK10 in PRAD by analyzing the RNA-seq data obtained from the TCGA database, and found that STK10 was abundantly expressed in 24 kinds of immune cells infiltrated in PRAD samples, especially in the T cells, the pivotal effector in the anti-tumor response (Figure 1B). Moreover, STK10 was significantly associated with stromal scores in PRAD, and STK10 was highly expressed in the adjacent normal tissues (Figure 1C,D). These data strongly suggest that STK10 may be involved in the progression of immune cell infiltration in prostate cancer.

### 3.2. Generation of Stk10-Deficient Mice

We first confirmed the tissue expression profile of Stk10 in mice by using qRT-PCR. cDNAs were prepared from 10 mouse tissues. Unique transcripts of mouse *Stk10* were detected dominantly in the immune tissues, including the spleen and bone marrow (Figure 2A). To further elucidate the role of Stk10 in the TME, we developed *Stk10* knockout mice by using CRISPR Cas9 technology. Two sgRNAs targeting the fourth exon were designed and then co-injected with Cas9 into mouse zygotes, and a mutant mouse line with a *Stk10* allele containing a 22 bp deletion in exon 4 was subsequently established (Figure 2B,C). With this deletion, the stop codon (TGA) appeared in frame at the position coding Leu145, resulting in the termination of translation. The absence of normal Stk10 protein was confirmed by a Western blot (Figure 2D). Moreover, the IHC results also demonstrated that mouse Stk10 is highly expressed in the spleens and some cells in the follicle of lymph nodes of WT mice, which is completely depleted in *Stk10*^−/−^ mice (Figure 2E).

### 3.3. RM-1 Tumor Growth Was Promoted in Stk10 Knockout Mice

To investigate whether host Stk10 deficiency alters tumor growth in mice, WT and *Stk10*^−/−^ mice were injected with RM-1-luciferase murine prostate cancer cells and we monitored the tumor growth. As shown in Figure 3, the growth rate of tumors in *Stk10*^−/−^ mice was faster than that in the WT mice. Accordingly, we found that the tumor weight and the tumor/body weight ratio were higher in *Stk10*^−/−^ mice. To explore the potential underlying mechanism of increased tumor burden in *Stk10*^−/−^ mice, we then examined the proliferation and apoptosis levels of tumor tissues. Cell proliferation was detected using IHC staining for Ki-67. The results showed that lack of host Stk10 results in accelerated proliferation of the tumor xenografts (Figure 4 ). On the contrary, the relatively lower apoptosis level of tumor xenografts in *Stk10*^−/−^ mice was determined by decreased expression of cleaved caspase 3 (Figure 4). These results indicated host *Stk10* depletion in mice accelerated tumor growth, suggesting the role of Stk10 as a tumor suppressor in vivo.

### 3.4. Stk10^−/−^ Mice Show Tumors with Decreased Infiltration of Activated and Effector CTLs

To analyze whether the accelerated tumor growth in *Stk10*^−/−^ mice is caused by TME changes, we analyzed the type and quantity of infiltrating immune cells in tumors from *Stk10*^−/−^ and WT mice 17 days after the injection of RM-1 cells. Flow cytometry analysis showed the percentage of total viable CD45^+^ infiltrated immune cells was comparable between tumors from *Stk10*^−/−^ and WT mice. We further evaluated the effect of *Stk10* knockout on the main leukocyte populations in the TME. The results showed that the targeted deletion of host Stk10 does not affect the percentage of tumor-infiltrated neutrophile granulocytes (CD45^+^CD11b^+^Gr-1^+^), macrophages (CD45^+^CD11b^+^F4/80^+^), NK cells (CD45^+^NK1.1^+^), NKT cells (CD45^+^NK1.1^+^CD3^+^) and B cells (B220^+^). However, the percentage of CD3^+^ cells was increased in the tumors grown from *Stk10*^−/−^ mice, suggesting more tumor-infiltrated T lymphocytes in the Stk10-deficient TME (Figure 5A–C). Further analysis revealed that CD4^+^ helper T cells (CD45^+^CD3^+^CD4^+^) were not significantly different between these two groups, and the percentage of tumor-infiltrated CD8^+^ CTLs (CD45^+^CD3^+^CD8^+^) was significantly higher in the *Stk10*^−/−^ mice compared to that in WT mice (Figure 5D). The percentages of apoptotic CTLs were comparable between these two groups (Appendix A).

CD8^+^ CTLs play important roles in the immune surveillance, clearance and immunotherapy of malignant tumors because they can directly kill cancer cells. Since our data revealed that knockout of host *Stk10* does not affect the cell apoptosis of CTLs, we next asked whether Stk10 participates in the activation of CTLs in response to tumors, because CTLs need to be activated in order to make efficient and durable anti-tumor immune responses. After examination of the main CTL subgroups, we found that *Stk10* deletion led to a significant increase in naïve CTLs (CD45^+^CD3^+^CD8^+^CD44^−^CD62L^+^), but decreases in activated (CD45^+^CD3^+^CD8^+^CD69^+^), effector (CD45^+^CD3^+^CD8^+^CD44^+^CD62L^−^) and exhausted CTLs (CD45^+^CD3^+^CD8^+^PD-1^+^LAG3^+^) (Figure 6). Collectively, our data provide some meaningful evidence that Stk10 may be critical for the activation and/or migration of CTLs.

### 3.5. Stk10 Knockout Induces Angiogenesis in the Tumor Tissue

Aberrant vasculature is one of the most common characteristics of malignant tumors. Numerous genes have been demonstrated to be associated in tumorigenesis by regulating angiogenesis in the tumor tissues. To uncover the potential contribution of Stk10 to tumor angiogenesis, we examined the expression of CD31, a marker of vascular endothelial cells, in the tumor tissues, by using immunofluorescence analysis. We found increased CD31 expression in the tumor tissues grown in *Stk10* knockout mice compared to those in the WT mice, which indicated that a lack of host Stk10 results in higher angiogenesis in the TME (Figure 7A). We further verified the role of Stk10 in the tumor vascular progression with a tube formation assay. The results showed that the average number of complete tubular structures significantly increased in the HUVECs cultured with the conditioned medium derived from *Stk10* knockout RM-1 cells (Figure 7B). These data suggest that Stk10 might play a suppressive role in tumor angiogenesis.

## 4. Discussion

The tumor microenvironment is a highly complex ecological niche comprised of various cell populations, extracellular constituents and a vascular network [13,17]. There are constant interactions and influences between these components, which support the tumor cells and help with migration. It is well documented that the TME not only participates in the progression of cancer growth and evolution, but also affects the therapeutic efficacy of numerous tumors [18,19,20]. In the past several years, more and more researchers have begun to focus on therapeutic approaches that target the tumor microenvironment [21,22]. Thus, identification of novel modulators of TME is necessary for the development of new therapeutic strategies for malignant cancers. Here, we report that targeted deletion of Stk10, a serine/threonine kinase, results in an aberrant TME in mice. In the current study, we investigated the function of host Stk10 in the tumorigenesis by using a syngeneic RM-1 murine prostate cancer cell line. Our data showed that lack of host Stk10 significantly potentiated RM-1 tumor growth in vivo. Increased tumor cell proliferation and decreased apoptosis caused by host Stk10 deletion are responsible for this phenotype.

It is well established that the immune cells enriched in the TME play pivotal roles in tumor growth and metastasis [23]. Previous reports and our data in this study demonstrated that STK10 is mainly expressed in the immune organs, such as the spleen and bone marrow [24]. Interestingly, IHC results showed that Stk10 expression in lymph nodes was concentrated in a small subset of cells in the follicle, further hints at the importance of Stk10 in the immune response. Most importantly, we found that STK10 was significantly associated with the tumor-infiltrating immune cells by using bioinformatic analysis. All this evidence gave us a clue that STK10 is involved in tumor growth by influencing the host’s anti-tumor immune response. To further verify this speculation, we evaluated the main populations of tumor-infiltrating immune cells. The FACS results showed that the proportion of naïve CTLs was significantly increased, and the proportions of activated, effector, and even exhausted CTLs were significantly decreased in tumor tissues grown in *Stk10*^−/−^ mice than those in the WT mice. The percentages of other leukocyte cell populations were not comparable in these two groups. As the key mediators of cellular immunity in response to tumors, naïve CTLs get activated upon the recognition of immunogenic antigen during tumorigenesis, and subsequently differentiate into effector CTLs which could specifically kill the tumor cells. Our data indicate that a lack of host Stk10 leads to a significant decrease in activated/effector CTLs and a relative increase in naïve CTLs infiltrated in the tumor tissues, suggesting that Stk10 may be involved in the activation of CTLs and/or participate in the migration of different CTL subsets in a cell specific manner.

Abundant angiogenesis in the tumor tissues is another typical characteristic of malignant tumors. The vascular system not only supplies nutrition and oxygen for tumor growth, but also presents a barrier to treatment via forming an immune-hostile microenvironment [11,25,26]. Notably, the current study demonstrated that the depletion of host Stk10 leads to a significant enhancement in vasculature formation in the tumor issues, as evidenced by CD31 staining. An in vitro assay further confirmed that Stk10 deficiency results in increased tuber formation. These results provided evidence that Stk10 might act as a suppressor of tumor angiogenesis. More importantly, differences in the vasculatures of tumor tissues grown from different genotypes of mice may also affect the migration of CTL subsets.

As well-established substrates of STK10, ezrin, radixin and moesin proteins act as mediators between the cytoskeleton and the plasma membrane. A number of studies have revealed that only phosphorylated ezrin and moesin are expressed in CTLs, localize within the microvilli, serve as the bridge between the actin cytoskeleton and TCRαβ complex, and participate in T cell activation upon antigen stimulation—during which there is a dynamic transformation between phosphorylation and dephosphorylation of many proteins [27,28]. Moreover, the role of ERM proteins in the angiogenesis of inflammatory diseases and cancers has been reported by different research groups [29,30,31]. However, whether the decreased tumor-infiltrated activated CTLs and increased tumor vasculature caused by host Stk10 deletion are mediated by the ERM protein remains to be further elucidated. To our knowledge, our work is the first attempt to explore the role of STK10 in the host’s anti-tumor immune response.

Taken together, target deletion of host *Stk10* resulted in uncontrolled tumor growth in mice, due to the deficiency of tumor-infiltrated activated/effector CTLs and abundant angiogenesis. Although further studies are needed to fully dissect the underlying molecular mechanism, our work provides strong evidence that Stk10 plays critical roles in the modulation of the tumor microenvironment and anti-tumor immune responses. Our work provides us with the possibility of establishing Stk10 as a candidate target for anti-tumor immunotherapy. Thus, we think that the manipulation of host Stk10 might be applicable in the management of cancer progression.

## Figures and Tables

**Figure 1 biology-11-01668-f001:**
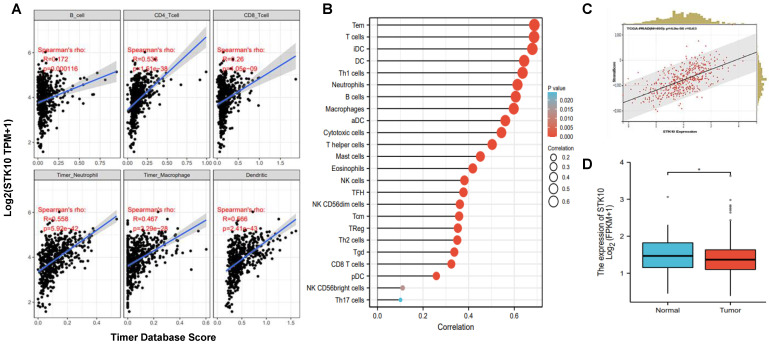
STK10 was associated with the infiltrated immune cells in prostate cancer. (**A**) The expression of STK10 showed significant positive correlations with B cells, CD4^+^ T cells, CD8^+^ T cells, neutrophils, macrophages and dendritic cells in the TME of PRAD. (**B**) Lollipop plots show the correlations between the STK10 expression level and the relative abundances of 24 types of immune cells in PRAD samples from the TCGA dataset. (**C**) The expression correlation of STK10 with stromal score in PRAD samples from the TCGA dataset. (**D**) The expression level of STK10 was significantly up-regulated in adjacent normal tissues compared to that in the tumor tissues. * *p* < 0.05.

**Figure 2 biology-11-01668-f002:**
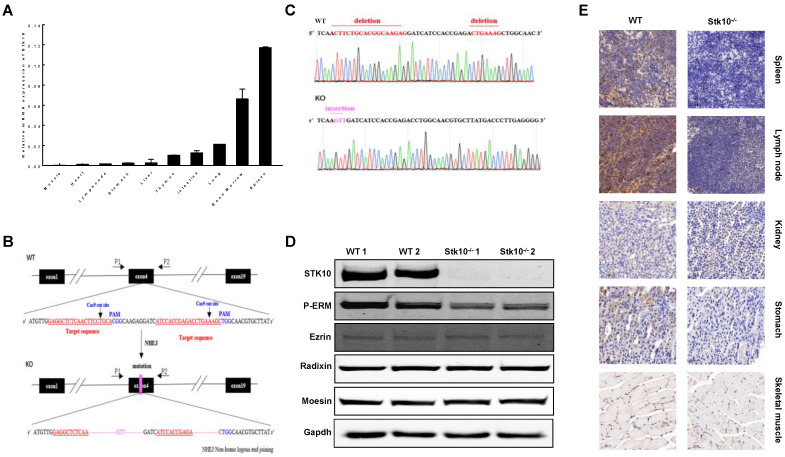
Construction of *Stk10* knockout mice. (**A**) Tissue expression profile of *Stk10* in C57BL/6J mice, with β-actin as an internal reference and expressed as mean ± standard deviation (*n* = 3/group). (**B**) Mouse *Stk10* knockout strategy; gRNA is the target sequence; NHEJ: non-homologous terminal connection. Wild type sequence is in black; Target sequence is in red; PAM sequence is in blue; Purple represents mutatant part in exon 4. (**C**) DNA sequencing map of *Stk10* knockout mice, showing a 22 bp deletion. Wild type sequence is in black; Deletion sequence is in red; Insertion sequence is in purple. (**D**) *Stk10* knockout mice were verified by Western blotting (WB) to have complete deletion of Stk10 protein and reduced phosphorylation levels of ERM proteins. (**E**) Expression levels of Stk10 in 5 tissues in *Stk10* knockout and WT C57BL/6J mice were detected by immunohistochemical staining. Scale bars, 50 μm.

**Figure 3 biology-11-01668-f003:**
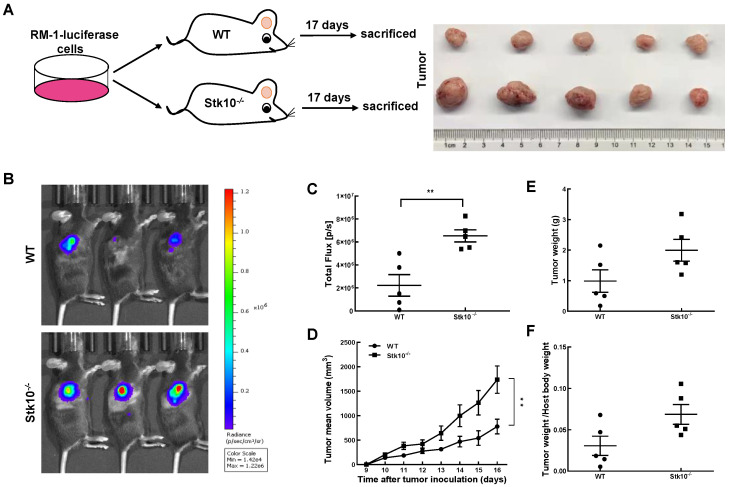
Deletion of *Stk10* promoted RM-1-luc tumor growth in C57BL/6J mice. RM-1-luc murine prostate cancer cells were directly injected into WT and *Stk10* knockout mice (*n* = 5/group). (**A**) A schematic diagram of the tumor-bearing mouse model and typical tumor photograph. (**B**) Detection of RM-1-luc tumors by in vivo fluorescence imaging. (**C**) Tumor luminescence intensity. (**D**) Tumor volume. (**E**) Tumor weight. (**F**) Ratio of tumor weight to host body weight. The results are represented as mean ± SEM from three independent experiments. Comparisons between groups were determined by Student’s *t*-test. ** *p* < 0.01.

**Figure 4 biology-11-01668-f004:**
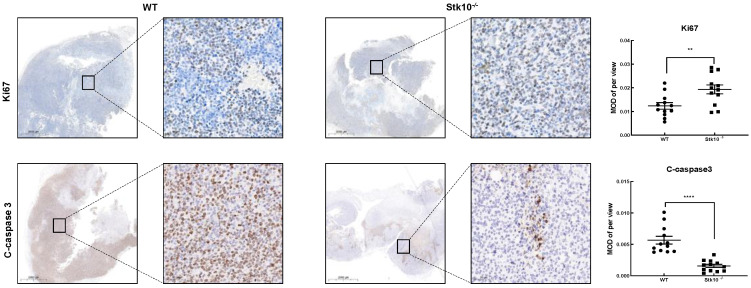
Stk10 deletion affected the proliferation and apoptosis of RM-1 tumor cells. Cell proliferation and apoptosis in the tumor tissue were analyzed by a immunohistochemistry assay for Ki67 and cleaved caspase-3, respectively (*n* = 3/group). Scale bars: 2000μm (**left**) and 50μm (**right**). The average optical density after Ki67 and cleaved caspase-3 staining was measured with Image Pro Plus 6.0 software (Media Cybernetics, Inc.). Data represent the mean ± SEM. MOD: mean optical density. ** *p* < 0.01; **** *p* < 0.0001.

**Figure 5 biology-11-01668-f005:**
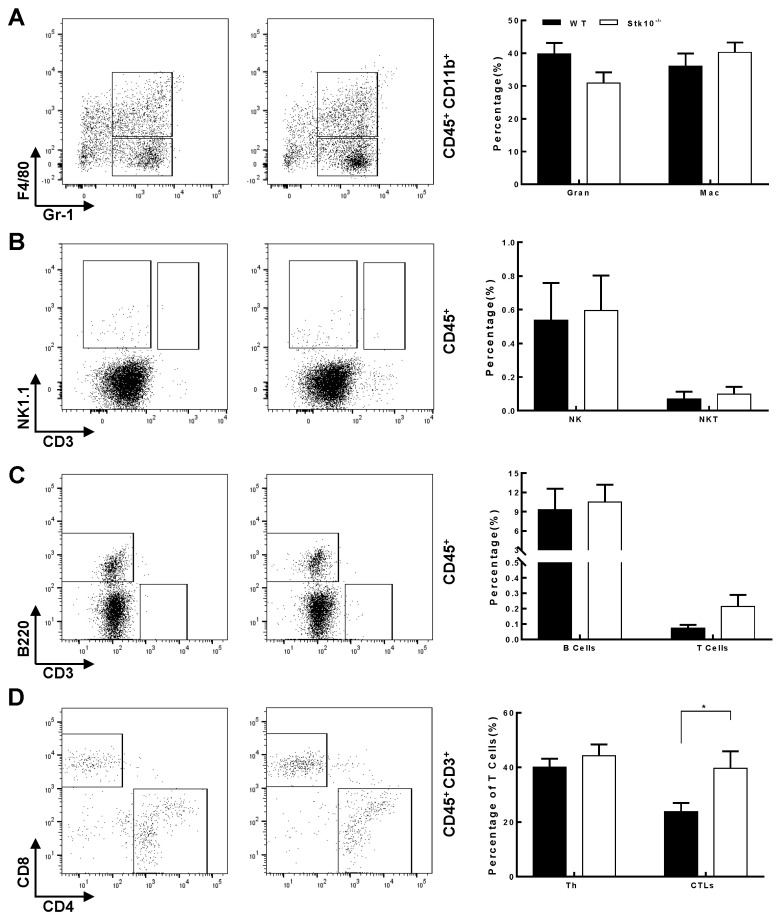
Flow cytometry analysis of tumor-infiltrating immune cells in tumors in WT and *Stk10* knockout mice. (**A**) Neutrophile granulocytes (CD45^+^CD11b^+^Gr-1^+^) and macrophages (CD45^+^CD11b^+^F4/80^+^). (**B**) NK cells (CD45^+^NK1.1^+^) and NKT cells (CD45^+^NK1.1^+^CD3^+^). (**C**) B cells (CD45^+^B220^+^) and T lymphocytes (CD45^+^CD3^+^). (**D**) Th cells (CD45^+^CD3^+^CD4^+^) and CTLs (CD45^+^CD3^+^CD8^+^). Representative results from three independent experiments are shown. Data represent the mean ± SEM. Statistical significance was determined by two-sided unpaired Student’s *t*-test. * *p* < 0.05. *n* = 5/group.

**Figure 6 biology-11-01668-f006:**
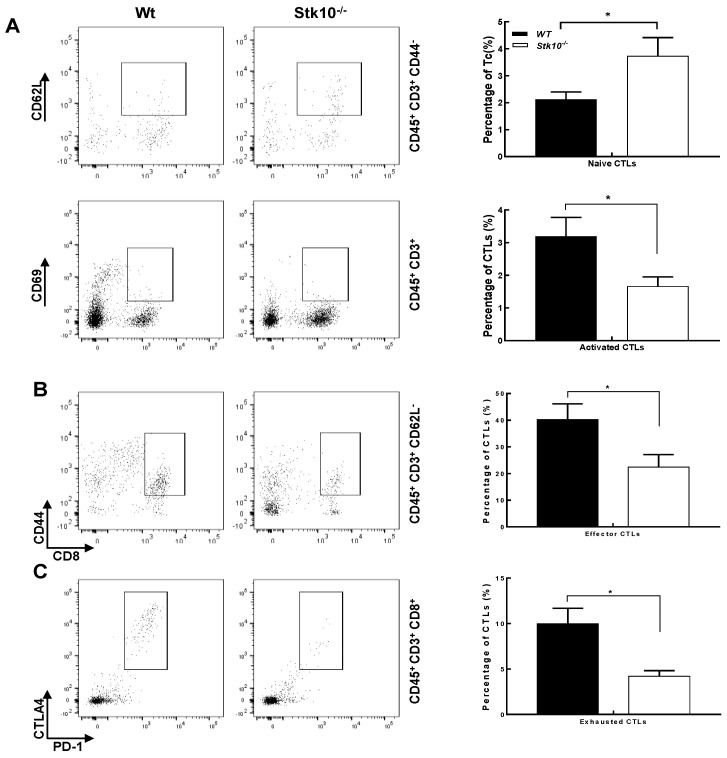
*Stk10* deletion led to a significant impairment in the activation of CD8^+^ T cells. (**A**) Naïve CTLs (CD45^+^CD3^+^CD8^+^CD44^–^CD62L^+^). (**B**) Activated CTLs (CD45^+^CD3^+^CD8^+^CD69^+^), effector CTLs (CD45^+^CD3^+^CD8^+^CD44^+^CD62L^−^). (**C**) Exhausted CTLs (CD45^+^CD3^+^CD8^+^PD-1^+^LAG3^+^). The frequencies (%) of CTLs at different stages in TILs cells are presented in the bar graphs as mean ± SEM. *p* values were determined by unpaired two-tailed *t*-tests. * *p* < 0.05. *n* = 5/group.

**Figure 7 biology-11-01668-f007:**
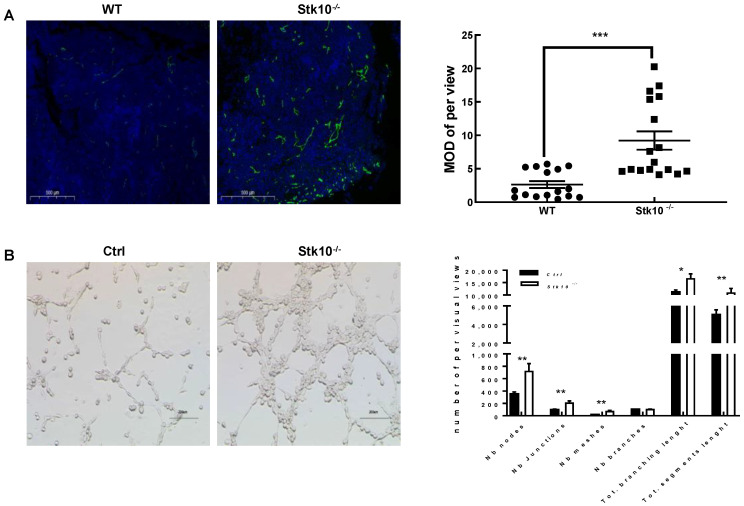
Stk10 deficiency led to abundant tumor angiogenesis. (**A**) CD31 expression on blood vessels in the RM-1 tumor tissues from WT and *Stk10*^−/−^ mice (*n* = 3/group). Scale bars: 500 μm. The average optical density of CD31 staining was measured with Image J Pro Plus 6.0 software (Media Cybernetics, Inc.). MOD: mean optical density. (**B**) Deletion of *Stk10* in RM-1cells inhibited angiogenic properties in HUVEC cells in vitro. The tubes were photographed under a microscope at 10× magnification. The tube nodes, branches, junction numbers and length were measured using the Image J software (version 1.8.0.112) The data are presented as relative means compared with control treatment. Data are presented as mean ± SEM of at least three independent experiments. * *p* < 0.05; ** *p* < 0.01; *** *p* < 0.001.

## Data Availability

The data presented in this study are available on request from the corresponding author. The data are not publicly available due to privacy.

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
