# Peer review of "Stk10 Deficiency in Mice Promotes Tumor Growth by Dysregulating the Tumor Microenvironment"

_biology, 2022, doi:10.3390/biology11111668_

Round 1

Reviewer 2 Report

Paper abstract appears interesting, however I cannot review it in its current format because the main figures seem very pixelated, and I am not able to visualize anything clearly to comment on them. Supplementary figures are of good quality, clear and can be visualized. Similarly, please resubmit the manuscript with clear and readable main figures so that it can be reviewed. 

Reviewer 3 Report

The research article by Jin-xin et al., describes the role of stk10 in tumor growth and metastasis in prostate cancer. In this article, authors used a stk10 deficient mice and demonstrated that stk10 is involved in immune cell infiltration and its deficiency led to tumor growth due to dysregulation of tumor microenvironment. Authors also demonstrated that stk10 deficiency led to increased angiogenesis. However, the manuscript needs to undergo revision before getting published.

Here are my comments:

1.       All the procedures and bioinformatic analysis performed by the authors were well described.

2.       Figures are blurred. Authors might want to provide original figures rather pasting them as pictures. Include the number of animals in every figure legend.

3.       A representative dot plot for the CTLs (Fig 5D) is preferred. It is difficult to observe the changes in the present one.

4.       Discussion section need to be improved. Add literature-based findings of Stk10 and CTLs and avoid repetition of results in the discussion section. Emphasize the significance of the study at the end of discussion.

Round 2

Reviewer 2 Report

This study identifies a previously unknown role for Stk10 in cancer progression, overall I think it is relevant and interesting, I only have minor points. 

1. Simple summary some minor work to improve language and sentence framing. "STK10 is significantly associated with tumor infiltrated immune cells" is vague, how are thy associated significantly or what it means is not clear.

2. Page 3, line 17-18- Why is the hyperlink to access protein atlas of STK10 included here, no explanation is given. 

3. Page 5, line 8- I would rephrase as STK10 knockout mice were "developed" at instead of "personally constructed".

4. In figure S1 legend, please mention what denotes Stk10, it is not clear currently what are in the figure shows Stk10 expression profile. 

5. Page 12, line 19-20, data discussed in these lines refer to Figure 2A, not Figure 1A. Please correct it.

6. In figure 2A, y axis title is written "Relationship of Stk10". I am not sure what it means. Did you mean "Relative expression of Stk10"? Please clarify.

7. Result section 3.3 needs minor English edits to improve language.
